# Chronoradiation Therapy for Prostate Cancer: Morning Proton Beam Therapy Ameliorates Worsening Lower Urinary Tract Symptoms

**DOI:** 10.3390/jcm9072263

**Published:** 2020-07-16

**Authors:** Hiromitsu Negoro, Takashi Iizumi, Yutaro Mori, Yoshitaka Matsumoto, Ichiro Chihara, Akio Hoshi, Hideyuki Sakurai, Hiroyuki Nishiyama, Hitoshi Ishikawa

**Affiliations:** 1Department of Urology, University of Tsukuba, 1-1-1 Tennodai, Tsukuba, Ibaraki 305-3575, Japan; chihara-tuk@umin.ac.jp (I.C.); a-hoshi@md.tsukuba.ac.jp (A.H.); nishiuro@md.tsukuba.ac.jp (H.N.); 2Department of Radiation Oncology, University of Tsukuba, Ibaraki 305-8575, Japan; iizumi@pmrc.tsukuba.ac.jp (T.I.); ymori@md.tsukuba.ac.jp (Y.M.); ymatsumoto@pmrc.tsukuba.ac.jp (Y.M.); hsakurai@pmrc.tsukuba.ac.jp (H.S.); hishikawa@pmrc.tsukuba.ac.jp (H.I.)

**Keywords:** radiation, timing, LUTS, circadian

## Abstract

Background and Purpose: Worsening lower urinary tract symptoms (LUTS) are a frequent adverse event following proton beam therapy (PBT) for localized prostate cancer. We investigated the differences in worsening LUTS among patients who received PBT at different times of day. Participants and Methods: Among 173 patients who underwent PBT for prostate cancer, 168 patients (median age 68.5 years) completed international prostate symptom score (IPSS) questionnaires and were included. Changes in the IPSS from baseline to the end of PBT were assessed by multiple linear regression analysis for age, National Comprehensive Cancer Network risk classification, androgen deprivation therapy, fractional PBT dose, clinical target volume, severity of IPSS, diabetes, LUTS medication use before PBT, anti-coagulant therapy and radiation time of day (morning (08:30–10:30), around noon (10:31–14:30), and late afternoon (14:31–16:30)). Results: IPSS total score and IPSS-Quality of Life (QoL) score (12 patients were excluded due to missing IPSS-QoL score) increased from eight to 14.9 (*p* < 0.0001) and from two to four (*p* < 0.0001), respectively. Time of day (morning) was the only determinant for worsening LUTS (β = −0.24, *p* < 0.01), voiding subscore (β = −0.22, *p* < 0.05) and IPSS-QoL (β = −0.27, *p* < 0.005), and was a determinant in item four (urgency) (β = −0.28, *p* < 0.005) with age (β = 0.19, *p* < 0.05). Conclusions: Morning PBT for localized prostate cancer significantly ameliorated worsening LUTS and improved QoL compared with treatment around noon or late afternoon. Chronoradiation therapy for localized prostate cancer may be effective and further research to elucidate the underlying mechanism is warranted.

## 1. Introduction

Proton beam therapy (PBT) is a promising curative therapy for localized prostate cancer as well as surgical resection and other radiation therapies (RTs) such as intensity-modulated radiation therapy and heavy particle therapy [1]. In Japan, national health insurance has covered PBT since April 2018 and the number of treatment facilities has increased. PBT for prostate cancer reduces the incidence of late toxicity because of smaller radiation doses and volumes directed at the organs at risk surrounding the prostate [1]. However, one of the relatively frequent adverse events following PBT treatment is lower urinary tract symptoms (LUTS). This significantly decreases the Quality of Life (QoL) of the patients. In a phase III clinical trial (PROG 9509), approximately half of the patients experienced grade 2 acute genitourinary morbidity based on Radiation Therapy Oncology Group criteria [2]. Worsening LUTS caused by PBT or other RTs result from inflammation or the acute reaction of the prostate, bladder neck or urethra. These symptoms are difficult to treat, despite the availability of non-steroidal anti-inflammatory drugs, alpha blockers, phosphodiesterase-5 inhibitors, anti-cholinergics, beta-3 agonists, and phytotherapeutic agents [3,4,5]. This represents a significant obstacle to the effective radiation treatment of prostate cancer. 

Chronotherapy is a treatment modality that utilizes circadian rhythms to optimize antitumor activity while minimizing adverse effects on healthy tissue [6,7]. Humans have a circadian clock system that controls cellular, physiological, metabolic and behavioral processes [8,9]. A circadian clock system exists in most tissues and cells, including the prostate [10], bladder [11], and immune system [12]. Therefore, any reaction to therapy may be influenced, in part, by the treatment time of day [6,7]. 

Chronoradiation is a relatively new concept from the perspective of circadian rhythm [6,7]. Similar to chronomodulated chemotherapy for cancer, selecting an optimal time of day for radiation treatment may result in a reduction in the inevitable side effects on surrounding healthy tissue, while increasing or maintaining therapeutic efficacy. At least 13 studies of chronoradiation have been reported and nine of these have demonstrated a significant difference in side effects, local tumor control, and overall survival based on time of treatment [13]. However, the number of studies performed remain inadequate and the evidence is still somewhat controversial. In the present study, we show that LUTS caused by PBT for localized prostate cancer may be influenced by time of day of treatment. 

## 2. Patients and Methods 

Between January 2013 and July 2018, PBT was performed in 173 patients with localized prostate cancer at the University of Tsukuba Hospital, Japan. Of these, 168 were included in this study and five patients were excluded as they did not complete the international prostate symptom score (IPSS) questionnaire. No patients had a previous surgical history of transurethral resection of the prostate. The baseline IPSS was obtained from patients prior to PBT and again during the last three sessions of PBT. Patients were staged according to TNM staging at clinical diagnosis using the National Comprehensive Cancer Network (NCCN) risk criteria. The clinical target volume (CTV) was set as the prostate plus 1/3 caudal seminal vesicle (whole seminal vesicle for cT3b). The planning target volume was defined as the CTV plus a 10-mm lateral, 12-mm anterior, and 5-mm craniocaudal and posterior margins [14]. Patients received 78 Gy in 39 fractions or 70 Gy in 28 fractions. The radiation time on the first day, corresponding to one fraction of total 39 or 28 fractions, was not fixed because an initial orientation was needed. On the 2nd day of treatment, patients were scheduled to receive PBT at a specific time of day between 8:30 to 16:30. Treatments were repeated at the same time depending on the patient’s preference or availability of the facility. In general, low risk patients did not receive androgen deprivation therapy (ADT), while intermediate and high-risk patients received neoadjuvant ADT for six months. High risk patients continued ADT as an adjuvant therapy for a total of three years. Medication for LUTS included alpha blockers, phosphodiesterase-5 inhibitors, anti-cholinergics, beta-3 agonists, and phytotherapeutic agents. Diabetes was assessed based on patient medical history or current use of antidiabetic therapy. Data were collected prospectively and were analyzed retrospectively based on time of day when PBT was performed. All patients provided written informed consent to allow their data to be used, and this study was approved by the Institutional Review Board at the University of Tsukuba Hospital (H29-135). 

## 3. Statistical Analyses

Significant differences in patients’ characteristics between the three groups were analyzed by the Kruskal–Wallis test or Fisher’s exact test. Factors significantly associated with changes in IPSS score from the baseline to the end of PBT were assessed using multiple linear regression analysis. These factors included age, NCCN risk classification, ADT, fractional PBT dose, clinical target volume (CTV), severity of IPSS, LUTS medication use before PBT, diabetes, anti-coagulant therapy and RT time of day. All statistical analyses were performed using the commercially available software package JMP 14.0.0 (SAS, Cary, NC, USA).

## 4. Results

The median age of the 168 enrolled patients treated with PBT in the morning (08:30–10:30), around noon (10:31–14:30), and late afternoon (14:31–16:30) was 68, 67 and 68.5 years, respectively. There was no significant difference in initial prostate-specific antigen (PSA) level, Gleason score, clinical tumor stage, NCCN risk classification, ADT, fractional PBT dose, CTV, IPSS total score, severity of IPSS, LUTS medication use before PBT, diabetes, or anti-coagulant therapy among these three groups (Table 1). 

During PBT, LUTS medications were added or changed by the decision of the radiologists or urologists based on the complaint of patients. The incidence tended to be lower in the morning group 7/52 (13.6%) compared with that in the around noon group 14/64 (21.9%) or that in the late afternoon group 15/52 (28.9%), which was not statistically significant (*p* = 0.16). No patients received antibiotics or nonsteroidal anti-inflammatory drug for LUTS during PBT. Following PBT, IPSS total score increased from 8 (range: 0–24) to 14.9 (range: 0–35, *p* < 0.0001), and IPSS-QoL score increased from 2 (range: 0–6) to 4 (range: 0–6, *p* < 0.0001). Table 2 summarizes the mean changes of IPSS scores between pre- and post-radiation and shows the results of multiple linear regression analyses for the change of IPSS score compared with RT time of day. Time of day (morning) was the only significant determinant (β = −0.24, *p* < 0.01) compared with NCCN risk classification, ADT, fractional PBT dose CTV, severity of IPSS, LUTS medication before PBT, diabetes, and anti-coagulant therapy. Morning radiation significantly reduced LUTS compared with radiation performed around noon or late afternoon as determined by post-hoc analysis (*p* < 0.05 each, Figure 1). As for IPSS subscore, morning RT was the only determinant in the voiding subscore (β = −0.21, *p* < 0.05), and a determinant in item four (urgency) (β = −0.28, *p* < 0.005) with age (β = 0.18, *p* < 0.05). We performed the same analysis for IPSS-QoL in the 156 patients (12 were excluded due to missing IPSS-QoL score). Morning radiation was the only determinant of less worsening IPSS-QoL, as assessed by multiple linear regression analysis (β = −0.27, *p* < 0.005) and post-hoc analysis, compared with radiation performed around noon or late afternoon (*p* < 0.05 each, Figure 1). 

## 5. Discussion

In this study, worsening LUTS, especially voiding symptoms and urgency, was significantly ameliorated in patients who received PBT for localized prostate cancer in the morning (before 10:30 AM) compared with patients receiving treatment after 10:30 AM. This effect remained significant after adjusting for various factors considered to be associated with worsening LUTS following PBT. These include age, NCCN risk classification, ADT, fractional PBT dose, CTV, severity of IPSS, LUTS medication use before PBT, diabetes, and anti-coagulant therapy. Our findings suggest that morning radiation reduces worsening LUTS and improves QoL. Thus, there may be underlying circadian mechanisms that determine differences in worsening LUTS following RT of prostate cancer. Hus et al. [15] found differences in adverse events and outcome associated with circadian variations in patients treated with RT for localized prostate cancer. This retrospective paper demonstrated that high dose RT (78Gy) administered in the evening (after 17:00) was significantly associated with late gastrointestinal complications of grade 2 or higher (hazard ratio = 2.96) compared with daytime RT (before 17:00). Using propensity score-matched analysis, the 5-year biochemical free survival was worse in the evening group compared with the daytime group (72% vs. 85%, hazard ratio = 1.95, *n* = 154). This increase in adverse events observed in evening therapy is consistent with our findings of a reduction in worsening LUTS in morning therapy and may be dependent on circadian rhythm.

Humans have a diurnal rhythm that is generated by an internal biological clock and is also influenced by external environmental factors. One hypothesis for the protective effect of RT is that higher levels of antioxidants in our body, including melatonin [16,17], growth hormones [18] and dehydroepiandrosterone [19,20], are circulating more abundantly in the morning compared with later times in the day under the control of our body clock. In particular, melatonin is not only a potent antioxidant and anti-inflammatory agent by neutralizing different types of free radicals and pro-oxidant enzymes activated by irradiation, but it is also known to have some anti-cancer properties. Melatonin may protect normal tissue while sensitizing tumor cells to radiation [17]. 

Another hypothesis is that the circadian clock exerts a protective effect on local tissues and cells following irradiation. Peripheral clocks influence many cellular processes including the redox state of cells, cell cycle, apoptosis, DNA damage response and metabolism in both normal tissues and cancer cells [21,22,23]. Dakup et al [24] revealed that the circadian clock has a cardioprotective effect against irradiation through transcriptional regulation of DNA damage response genes in the heart. Johnson et al [25] reported that single nucleotide polymorphisms in clock genes are associated with the severity of toxicity of breast cancer radiotherapy at different times of day. Clock genes may be directly associated with the severity of adverse events at different treatment times. In addition, local clock-controlled genes, which are approximately 10% of the genes oscillated under the control of circadian rhythm [26], may influence the reaction to RT. Since their expression levels vary between organs and tissues [26], this tissue specificity could be related to the different reactions to RT time among organs. 

Our study has several limitations. First, the study design was retrospective and not randomized. Given that the time of day for PBT was fixed in all evaluated patients at our institution and only five patients were excluded from the analysis, the influence of selection bias was small. Furthermore, patients who underwent PBT did not receive any information from the hospital staff as to the possible differential effects of treatment time of day on adverse events to avoid any cognitive bias. Clinicians also prescribed medications for LUTS during PBT based on the complaints of patients, without having any bias related to the difference in worsening LUTS by the time of day of PBT, since this study was retrospectively conducted. The lower tendency to prescribe LUTS medication in the morning group suggested that patients in the morning group complained of LUTS less compared with those in other groups, and that the difference in worsening IPSS by the time of day was not caused by the effect of medications for LUTS. Second, LUTS severity may be influenced by prostate volume. We used CTV, which includes whole prostate and proximal seminal vesicles, instead of prostate volume as a variable for multivariate analysis. This was done because prostate volume in some patients was lost from the data in the treatment planning system. However, it appears that CTV may be a surrogate for the influence of prostate volume in the present study. Third, the mean difference in worsening IPSS in relation to the time of radiation therapy, 2.33 (morning vs. around noon) and 2.92 (morning vs. late afternoon), might be somewhat modest. There are some medications for LUTS such as alpha blockers, anti-cholinergics, beta-3 agonists, phytotherapeutic agents, and nonsteroidal anti-inflammatory drugs, but their effects could be somewhat insufficient for LUTS induced by irradiation. Given this current situation, the significant amelioration of worsening LUTS can be clinically meaningful. Fourth, objective examinations for LUTS were not performed with respect to uroflowmetry, the measurement of residual urine, urodynamics studies and voiding diaries. Voiding symptoms and urgency had significant differences with respect to time of day, and signs of bladder outlet obstruction and/or detrusor overactivity may be detected by such examinations. It is recommended that objective data are obtained when prospective studies are designed. 

## 6. Conclusions

Our data indicate that morning RT for localized prostate cancer may ameliorate worsening LUTS and improve QoL compared with treatment around noon or late afternoon. Although chronoradiotherapy for prostate cancer may be an effective way to reduce worsening LUTS, further research is needed to elucidate the mechanism of this effect. The identification of other optimal times of day to perform PBT will also need to be established and coordinated with the availability of clinical personnel and resources. 

## Figures and Tables

**Figure 1 jcm-09-02263-f001:**
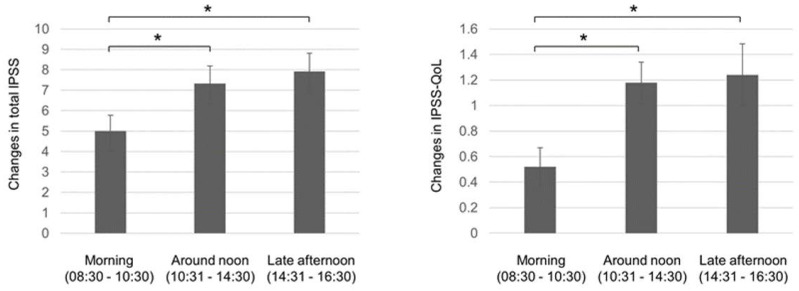
Differences in change of IPSS total score and IPSS-Quality of Life (QoL) score between pre- and post-proton beam therapy by time of day for radiation. IPSS, international prostate symptom score; Quality of Life (QoL). * *p* < 0.05 by the Dunnett test as a post-hoc analysis for multiple linear regression analysis.

**Table 1 jcm-09-02263-t001:** Characteristics of the 168 patients who received proton beam therapy for localized prostate cancer.

	Time of Day for Proton Beam Therapy				
	Morning	Around Noon	Late Afternoon		
	(08:30–10:30)	(10:31–14:30)	(14:31–16:30)	Total	*p* Value
No. patients (%)	52 (31.0)	64 (38.1)	52 (31.0)	168	
Age at proton therapy (median [range], years)	68 (57–80)	67 (56–86)	68.5 (53–86)	68 (53–86)	0.8
Initial PSA (median [range], ng/mL)	10.3 (3.5–220)	7.66 (3.5–143)	8.3 (1.6–150)	8.6 (1.6–220)	0.39
Gleason score at prostate biopsy (*n*, %)					0.37
6	10 (19.2)	13 (20.3)	6 (11.5)	29 (17.3)	
7	20 (38.5)	30 (46.9)	21 (40.4)	71 (42.3)	
8	15 (28.9)	15 (23.4)	13 (25)	43 (25.6)	
9	6 (11.5)	5 (7.8)	12 (23.1)	23 (13.7)	
10	1 (1.9)	1 (1.6)	0	2 (1.2)	
Clinical T stage (*n*, %)					0.77
T1	12 (23.1)	16 (25)	10 (19.2)	33 (22.6)	
T2	27 (51.9)	35 (54.7)	26 (50)	88 (52.4)	
T3	13 (25)	13 (20.3)	16 (30.8)	42 (25.0)	
NCCN risk classification (*n*, %)					0.67
Low	7 (13.5)	9 (14.1)	4 (7.7)	20 (11.9)	
Intermediate	20 (38.5)	26 (40.6)	18 (34.6)	64 (38.1)	
High	25 (48.1)	29 (45.3)	30 (57.7)	84 (50.0)	
ADT (*n*, %)	46 (88.5)	52 (81.3)	48 (92.3)	146 (86.9)	0.21
Fractional PBT dose (*n*, %)					0.47
2.0 Gy	25 (48.1)	24 (37.5)	24 (46.2)	73 (43.5)	
2.5 Gy	27 (51.9)	40 (62.5)	28 (53.9)	95 (56.5)	
CTV (median [range], mL)	29.8 (15.1–61.9)	29.5 (16.9–98.6)	28 (16.4–87.1)	29 (15–84)	0.7
IPSS total score (median [range])	8 (0–18)	8.5 (0–24)	8 (0–20)	8 (0–24)	0.84
Severity of IPSS (*n*, %)					0.44
Mild	24 (46.2)	26 (40.6)	25 (48.1)	75 (44.6)	
Moderate	28 (53.9)	34 (53.1)	26 (50)	88 (52.4)	
Severe	0 (0)	4 (6.3)	1 (1.9)	5 (3.0)	
IPSS-QoL score (median [range])	2 (0–5)	2 (0–6)	2 (0–6)	2 (0–6)	0.79
LUTS medication use before PBT (*n*, %)	3 (5.8)	8 (12.5)	3 (5.8)	14 (8.3)	0.31
Diabetes (*n*, %)	14 (26.9)	11 (17.2)	8 (15.4)	33 (19.6)	0.29
Anti-coagulant therapy (*n*, %)	9 (17.3)	8 (12.5)	8 (13.4)	25 (14.9)	0.8

Prostate-specific antigen (PSA); National Comprehensive Cancer Network (NCCN); androgen deprivation therapy (ADT); proton beam therapy (PBT); clinical target volume (CTV); international prostate symptom score (IPSS); lower urinary tract symptoms (LUTS).

**Table 2 jcm-09-02263-t002:** Changes in IPSS compared with time of day for radiation.

	Time of Day for Proton Beam Therapy				
	Morning	Around Noon	Late Afternoon		
	(08:30–10:30)	(10:31–14:30)	(14:31–16:30)	β	*p* Value
IPSS total score	5 ± 0.77	7,33 ± 0.86	7.92 ± 0.87	−0.24	0.008
IPSS voiding subscore	2.85 ± 0.55	4.12 ± 0.58	4.88 ± 0.66	−0.21	0.02
IPSS storage subscore	2.15 ± 0.39	3.14 ± 0.42	3.26 ± 0.43		n.s.
IPSS subscore					
1	0.56 ± 0.15	0.92 ± 0.17	0.87 ± 0.19		n.s.
2	0.85 ± 0.22	1.03 ± 0.20	1.08 ± 0.22		n.s.
3	0.81 ± 0.18	0.92 ± 0.19	1.27 ± 0.20		n.s.
4	0.52 ± 0.15	1.13 ± 0.18	1.22 ± 0.22	−0.28	0.002
5	0.98 ± 0.25	1.30 ± 0.21	1.73 ± 0.27		n.s.
6	0.5 ± 0.17	1.05 ± 0.16	0.94 ± 0.21		n.s.
7	0.79 ± 0.14	0.98 ± 0.18	0.96 ± 0.14		n.s.
IPSS-QoL score	0.52 ± 0.15	1.18 ± 0.16	1.24 ± 0.24	−0.27	0.004

International prostate symptom score (IPSS). Data were presented as the mean ± SEM. *p* < 0.05 was considered significant by multiple linear regression analysis.

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
