# Peer review of "Chronoradiation Therapy for Prostate Cancer: Morning Proton Beam Therapy Ameliorates Worsening Lower Urinary Tract Symptoms"

_jcm, 2020, doi:10.3390/jcm9072263_

Round 1

Reviewer 1 Report

This is a nice study investigating the impact of the time of the day proton treatment (morning, noon, afternoon) is received on the side effects experienced by prostate cancer patients.  with regression analysis, the authors identify that "Morning radiation was 113 the only determinant of less worsening IPSS-QoL".

There is a good rationale and possible mathematical determinant to carefully choosing the time of treatment (McLaughlin et al. 2014 - DOI: https://doi.org/10.1017/S1460396913000411 

It can not be excluded that a confounding factor may explain the difference seen in this study.  But the authors have carefully balanced their group for a number of relevant factors including anticoagulant therapy and diabetes. it is thus possible that morning irradiation best suits these patients. Further evaluation is warranted. 

Author Response

This is a nice study investigating the impact of the time of the day proton treatment (morning, noon, afternoon) is received on the side effects experienced by prostate cancer patients with regression analysis, the authors identify that "Morning radiation was the only determinant of less worsening IPSS-QoL".

There is a good rationale and possible mathematical determinant to carefully choosing the time of treatment (McLaughlin et al. 2014 - DOI: https://doi.org/10.1017/S1460396913000411)

It can not be excluded that a confounding factor may explain the difference seen in this study. But the authors have carefully balanced their group for a number of relevant factors including anticoagulant therapy and diabetes. it is thus possible that morning irradiation best suits these patients. Further evaluation is warranted.

Thank you very much for your positive comment for our article. We appreciate and agree with the comments of reviewer 1, and it is necessary to evaluate the conclusion of this study from both basic and clinical sciences. 

Reviewer 2 Report

Study Design: Was this retrospective or prospective? This was noted in the discussion but not mentioned in in the methods section.  Please include in the methods.

When was the IPSS questionnaire delivered at the end of treatment? Same day? Within 1-2 weeks? Please specify timing. 

Methods: Treatment volumes (prostate + seminal vesicles?, nodes for high-risk patients? CTV --> PTV expansions?) were briefly mentioned in the discussion but not the methods section. Please describe in methods section in more detail. 

Some important details were left out about the patients:

  • Prior TURP?
  • LUTS-directed therapy before, during and after treatment (tamsulosin, oxybutinin, pyridium, etc) -- THIS IS VERY IMPORTANT -- as this could significantly affect IPSS scores. It is hard to draw conclusions without this information.
  • Incidence of antimicrobial therapy for prostatitis / cysitits

Discussion: I am not sure I agree with this statement: "However, considering the current situation without any established medication for LUTS induced by 172 irradiation, the significant amelioration of worsening LUTS can be clinically meaningful." There are medications that help LUTS (alpha-adrenergic antagonists (tamsulosin), anticholinergic (oxybutinin), NSAIDs, etc). 

Author Response

Study Design: Was this retrospective or prospective? This was noted in the discussion but not mentioned in the methods section. Please include in the methods.

 Thank you for your comment. Data were collected prospectively, while this study was conducted retrospectively. We have added this in the methods section as below;

Line 86: “Data were collected prospectively and were analyzed retrospectively based on time of day when PBT was performed.”

When was the IPSS questionnaire delivered at the end of treatment? Same day? Within 1-2 weeks? Please specify timing.

The IPSS questionnaire was delivered in the final week of PBT at the outpatient clinic of radiologists, which was during the last three sessions of PBT. Therefore, we have modified the sentence as below;

From

The baseline IPSS was obtained from patients prior to PBT and again at the end of treatment

Line 72:

To

The baseline IPSS was obtained from patients prior to PBT and again during the last three sessions of PBT.

 Methods: Treatment volumes (prostate + seminal vesicles?, nodes for high-risk patients? CTV --> PTV expansions?) were briefly mentioned in the discussion but not the methods section. Please describe in methods section in more detail.

 In accordance with the reviewer’s suggestion, we have added the sentence and a reference as below;

Line 74: “The clinical target volume (CTV) was set as the prostate plus 1/3 caudal seminal vesicle (whole seminal vesicle for cT3b). The planning target volume was defined as the CTV plus a 10-mm lateral, 12-mm anterior, and 5-mm craniocaudal and posterior margins14.”

New Ref.14

Makishima H, Ishikawa H, Tanaka K, et al. A retrospective study of late adverse events in proton beam therapy for prostate cancer. Mol Clin Oncol. 2017; 7:547.

 Some important details were left out about the patients: Prior TURP? LUTS-directed therapy before, during and after treatment (tamsulosin, oxybutinin, pyridium, etc) -- THIS IS VERY IMPORTANT -- as this could significantly affect IPSS scores. It is hard to draw conclusions without this information. Incidence of antimicrobial therapy for prostatitis / cystitis

 Thank you for your valuable comments. No patients had a history of TUR-P. In accordance with the reviewer’s comments, we have added the information on medication use for LUTS before and during PBT. The medication for LUTS before PBT has been added in the logistic regression analysis and re-analyzed, while the results of the analysis got little different. The incidence of add or change of LUTS medication during PBT has been added in the result section. Since this study was focused on the acute worsening of LUTS, we did not add the information on medication after PBT. No patients received antimicrobial therapy for prostatitis/cystitis during PBT. We have added this information as below;

Line 71: “No patients had a previous surgical history of transurethral resection of the prostate.”

Line 84: “Medication for LUTS included alpha blockers, phosphodiesterase‐5 inhibitors, anti-cholinergics, beta-3 agonists, and phytotherapeutic agents.”

Line 26, 95, 102, 121 and 144: “LUTS medication use before PBT”

Line 111: During PBT, LUTS medications were added or changed by the decision of the radiologists or urologists based on the complaint of patients. The incidence tended to be lower in the morning group 7/52 (13.6%) compared with that in the around noon group 14/64 (21.9%) or that in the late afternoon group 15/52 (28.9%), which was not statistically significant (p=0.16). No patients received antibiotics or nonsteroidal anti-inflammatory drug for LUTS during PBT.

Line 181: Clinicians also prescribed medications for LUTS during PBT based on the complaint of patients without having any bias for difference for worsening LUTS by the time of day of PBT since this study was retrospectively conducted. The lower tendency of prescription for LUTS medication in the morning group suggested that patients in the morning group complained LUTS less compared with those in other groups, and that the difference of worsening IPSS by the time of day was not caused by the effect of medications for LUTS.

Discussion: I am not sure I agree with this statement: "However, considering the current situation without any established medication for LUTS induced by 172 irradiation, the significant amelioration of worsening LUTS can be clinically meaningful." There are medications that help LUTS (alpha-adrenergic antagonists (tamsulosin), anticholinergic (oxybutinin), NSAIDs, etc).

In accordance with the reviewer’s comment, we have modified the sentence as below;

From “However, considering the current situation without any established medication for LUTS induced by irradiation, the significant amelioration of worsening LUTS can be clinically meaningful.”

Line 190: To “There are some medications for LUTS such as alpha blockers, anti-cholinergics, beta-3 agonists, phytotherapeutic agents, and nonsteroidal anti-inflammatory drug, while their effects could be somewhat insufficient for LUTS induced by irradiation. Given this current situation, the significant amelioration of worsening LUTS can be clinically meaningful.”

We wish to express again our deep appreciation for all comments, which have helped us to improve the manuscript significantly.
